# Integrating isoniazid preventive therapy into the fast-track HIV treatment model in urban Zambia: A proof-of -concept pilot project

**Mpande Mukumbwa-Mwenechanya**[1]*, **Muhau Mubiana**[1], **Paul Somwe**[1], **Khozya Zyambo**[2], **Maureen Simwenda**[3], **Nancy Zongwe**[2], **Estella Kalunkumya**[1], **Linah Kampilimba Mwango**[4], **Miriam Rabkin**[5,6], **Felton Mpesela**[7], **Fred Chungu**[8], **Felix Mwanza**[9], **Peter Preko**[10], **Carolyn Bolton-Moore**[1,11], **Samuel Bosomprah**[1,12], **Anjali Sharma**[1], **Khunga Morton**[2], **Prisca Kasonde**[13], **Lloyd Mulenga**[2], **Patrick Lingu**[2], **Priscilla Lumano Mulenga**[2]

1 Centre for Infectious Diseases Research in Zambia, Lusaka, Zambia, 2 Ministry of Health, Lusaka, Zambia, 3 JSI USAID SAFE, Lusaka, Zambia, 4 CIHEB, Lusaka, Zambia, 5 ICAP at Columbia University and Departments of Medicine & Epidemiology, New York, New York, United States of America, 6 Columbia University Mailman School of Public Health, New York, New York, United States of America, 7 Clinton Health Access, Lusaka, Zambia, 8 Network of Zambian People Living with HIV, Lusaka, Zambia, 9 Treatment Advocacy and Literacy Campaign, Lusaka, Zambia, 10 ICAP at Columbia University, Mbabane, Eswatini, 11 University of Alabama at Birmingham, Birmingham, Alabama, United States of America, 12 Department of Biostatistics, School of Public Health, University of Ghana, Accra, Ghana, 13 ICAP at Columbia University, Lusaka, Zambia

* Mpande.Mwenechanya@cidrz.org

**Data Availability Statement:** The Government of Zambia allows data sharing when applicable local

## Abstract

Most people living with HIV (PLHIV) established on treatment in Zambia receive multi-month prescribing and dispensing (MMSD) antiretroviral therapy (ART) and are enrolled in less-intensive differentiated service delivery (DSD) models such as Fast Track (FT), where clients collect ART every 3–6 months and make clinical visits every 6 months. In 2019, Zambia introduced Isoniazid Preventive Therapy (IPT) with scheduled visits at 2 weeks and 1, 3, and 6 months. Asynchronous IPT and HIV appointment schedules were inconvenient and not client centered. In response, we piloted integrated MMSD/IPT in FT HIV treatment model. We implemented and evaluated a proof-of-concept project at one purposively selected high-volume facility in Lusaka, Zambia between July 2019 and May 2020. We sensitized stakeholders, adapted training materials, standard operating procedures, and screened adults in FT for TB as per national guidelines. Participants received structured TB/IPT education, 6-month supply of isoniazid and ART, aligned 6th month IPT/MMSD clinic appointment, and phone appointments at 2 weeks and months 1–5 following IPT initiation. We used descriptive statistics to characterize IPT completion rates, phone appointment keeping, side effect frequency and Fisher's exact test to determine variation by participant characteristics. Key lessons learned were synthesized from monthly meeting notes. 1,167 clients were screened with 818 (70.1%) enrolled, two thirds (66%) were female and median age 42 years. 738 (90.2%) completed 6-month IPT course and 66 (8.1%) reported IPT-related side effects. 539 clients (65.9%) attended all 7 telephone appointments. There were insignificant differences of outcomes by age or sex. Lessons learnt included promoting

conditions are satisfied. In this case, the anonymised data and tools from the study have been made available and uploaded as supplementary information.

**Funding:** This work was supported in part, by the Bill & Melinda Gates Foundation [grant number OPP1152764]. Under the grant conditions of the Foundation, a Creative Commons Attribution 4.0 Generic License has already been assigned to the Author Accepted Manuscript version that might arise from this submission. The funder had no role in study design, data collection and analysis, decision to publish, or preparation of the manuscript. Furthermore, none of the authors received a salary from the funders.

**Competing interests:** The authors have declared that no competing interests exist.

project ownership, client empowerment, securing supply chain, adapting existing processes, and cultivating collaborative structured learning. Integrating multi-month dispensing and telephone follow up of IPT into the FT HIV treatment model is a promising approach to scaling-up TB preventive treatment among PLHIV, although limited by barriers to consistent phone access.

## Introduction

Globally, tuberculosis (TB) is the direct cause of one-third of all human immunodeficiency virus (HIV) related deaths and the leading cause of death among people living with HIV (PLHIV) [1]. PLHIV account for approximately 10% of the 10 million annual new TB cases and on average, have about 21-fold (CI: 16–27) higher risk of developing TB than HIV-negative persons [2–5]. Zambia has one of the highest TB burdens in sub-Saharan Africa (SSA) and is ranked among the top 30 TB control priority countries by the World Health Organization (WHO) [1]. In 2021, the estimated 59,000 new cases of TB in Zambia contributed to approximately 14,800 deaths among Zambians, of which 9,100 (62%) occurred among PLHIV [2, 3].

The WHO Global End TB strategy recommends provision of TB preventive treatment (TPT) for PLHIV with the aim of eliminating TB by 2030. Whilst a priority and cost-effective, TPT coverage and uptake has been low among PLHIV and current TPT coverage remains well under WHO's 2025 coverage target ($\geq$ 90%) [3, 6–10]. Commonly cited barriers for low uptake and coverage include stock-outs, perceived fears of poor adherence and isoniazid (INH) resistance, inability to prevent and monitor adverse events, a lack of local or national commitment, and perceived difficulty of ruling out active TB [11–13].

In 2019, the Zambian Ministry of Health (MoH) adopted the WHO-recommended TPT strategy [2]. The MoH TPT guidance provided for Isoniazid Preventive Therapy (IPT) dispensed at 2 weeks then at 1, 3 and 5 months for PLHIV established on treatment. PLHIV established on treatment defined as non-pregnant adults on ART for greater than 6 months, with VL less than 1000 copies/mL, and without opportunistic infections or active TB. This same population was eligible to enroll in Differentiated Service Delivery (DSD) for HIV treatment, where multi-month (3–6 months) scripting and dispensation (MMSD) of ART is provided [14, 15]. One prominent DSD model utilizing MMSD in Zambia is the Fast-Track (FT) model, an individual facility-based model in which clients visit health facilities every three months, alternating between expedited medication pick-up visits and clinical visits (Fig 1) [14]. By June 2021, 13% of people on ART in Zambia had enrolled in FT [15]. Whilst integration of IPT into this model is an important part of TPT scale up, the asynchronized appointment schedules for ART specific medication pick-up and IPT refills threaten DSD's "client centered" approach and thus retention in HIV-TB care.

Integrating multi-month dispensing of IPT into a rapidly growing DSD model for HIV care offers a promising public health strategy to improve TPT uptake and prevention of TB among PLHIV [16]. DSD models with longer refill intervals have demonstrated high rates of long-term retention where 90% remained in care at four years compared to less than 70% in traditional facility-based HIV programs with monthly visits [17–20]. While case findings from SSA countries have shown that integration of MMSD TPT into DSD models is feasible, the IPT refills were limited to a maximum of three months with minimal evidence of adherence support and side-effect monitoring at shorter intervals via phone [16]. Furthermore, the use of mobile phones for continuous PLHIV monitoring and adherence support and provision of 6MMSD TPT is silent in framework documents for programmatic considerations of TPT

| Fast Track Eligibility Criteria | Fast Track Processes |
|---|---|
| • Age > 18 years<br>• On ART > 12 months<br>• Viral load < 1,000 copies/mL<br>• No acute illness<br>• No complex comorbidities, opportunistic infections or active TB | • Health facility visits every 3 months for accelerated pharmacy pick up and adherence counseling<br>• Health facility visits every 6 months for full clinical visits + pharmacy visit + adherence counseling |

**Fig 1. The fast -track HIV treatment model.**

implementation in DSD models [21]. In response, we designed a proof-of-concept pilot project that integrated MMSD and IPT services in the FT DSD model by synchronizing long refills of IPT (6 months MMSD) and visit schedules and utilized phone calls to screen for TB symptoms, provide adherence support and side effect monitoring. The approach was piloted at a single health facility, given limited resources and the desire to start small as well as learn from experience. Objectives of study included documenting the inputs, processes, enabling factors and challenges involved in launching this new approach to TPT delivery, along with client uptake, completion of phone check-ins, self-reported TPT completion rates and side effects. We did not attempt to compare TPT outcomes *vs.* standard of care or to assess generalizability.

## Methods

### Study site

We purposively selected a high-volume HIV clinic serving more than 10,000 PLHIV active on ART (including 5,000 enrolled in FT) located within an urban primary level health centre in Lusaka. We believed this site to be relatively common in our country (even if not typical) to quickly learn and generate valuable lessons that could contribute to the dialogue about integration of TPT into DSD models.

### Study design and participants

The pilot was implemented in three phases: preparatory, enrollment, and intervention phase (Fig 2). Inclusion criteria included age over 18 years, being established on ART as per national guidelines, enrolled in FT, having access to a functioning phone, and willing to participate. Exclusion criteria included recent or current IPT use, recent or current TB treatment, current TB symptoms, and pregnancy. All clients that came to the facility during the one-month enrollment period (September 2019 –October 2019) were eligible to enroll.

We piloted the intervention, consisting of an adjusted IPT visit schedule; baseline education and counselling; dispensing a 6 month-supply of both IPT and ART; phone check-ins at 2

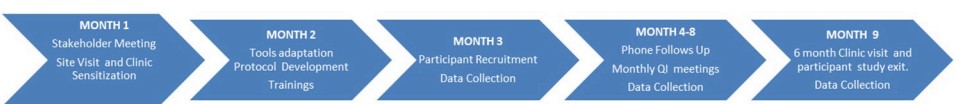

**Fig 2. Project implementation timeline.**

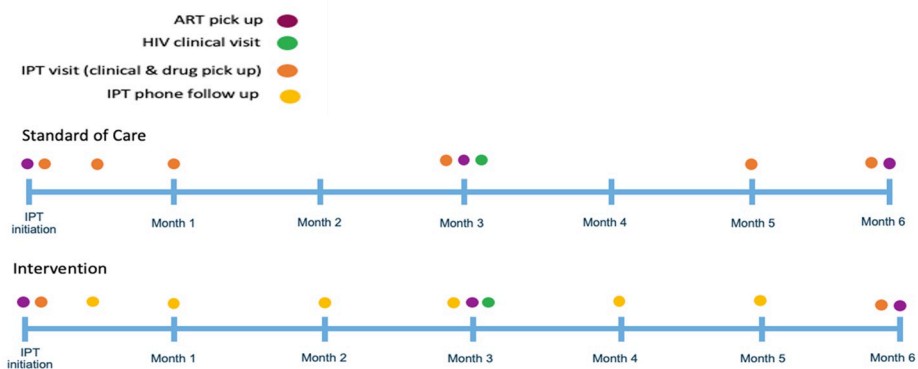

**Fig 3. Intervention vs. standard of care.**

weeks and then monthly for 5 months; and a synchronized in-person follow-up visit for HIV and IPT care at 6 months (Fig 3).

## Study procedures

Before protocol development, the pilot team carefully documented FT model processes and reviewed the existing supply chain and infrastructure. This information was used to design the intervention model (Fig 3) and to identify the appropriate facility staff to provide screening, counselling, enrolment, IPT initiation and follow-up services.

Cognizant that a strong supply chain system that ensures availability of both INH and Pyridoxine (Vitamin B6) was pivotal for project success, the project team engaged the NTP Pharmacist and health facility Pharmacist to secure a constant supply of commodities [22]. The project team developed, and adapted training materials, standard operating procedures (SOPs) and data collection tools and conducted stakeholder sensitization with Civil Society Organizations (CSOs), healthcare workers (HCWs) and PLHIV to ensure initial and ongoing success of IPT incorporation into FT. We conducted in-person didactic training, reoriented health centre staff to standardized screening for IPT eligibility and management of patients on IPT including symptom screening and counselling. Furthermore, community health workers (CHWs) were trained on conducting structured phone follow-ups and provide referral for clinical consultations.

During the month-long enrolment phase (September 2019—October 2019), HCWs routinely screened adults in FT model for pilot project eligibility. Eligible clients were invited to enroll in the integrated IPT/ART model, participants then received structured health education messages on the importance and benefits of IPT as well as identification, reporting and management of TB symptoms/expected side effects. Enrollees received 6 months dispensation of INH (300mg), pyridoxine (Vitamin B6), their current ART regimen, and an appointment to return in 6 months.

During the intervention phase, CHW conducted phone follow-ups at 2 weeks, 4 weeks, and then once monthly for 5 months. Structured checklist and script was utilized to assess TB symptoms, IPT adherence, any related side-effects and concerns/experiences (see Additional file, Section one). When participants were not available by phone after two failed attempts on two separate days, the CHWs followed up with physical visits to their homes using the address provided by the clients at enrollment. The study team shared the data at monthly meetings with all DSD staff at the facility to; review progress, identify challenges, facilitate the use of quality improvement methods to develop and test adjustments as required. We read and re-

read all monthly meeting notes to identify and abstract all text on lessons learned during implementation including contextual factors (enablers and hinderances) and attendees' reflections on their own actions and relationships. Notes from these meetings and discussions were used to synthesize key lessons learned, we classified the success as enabling factors and challenges as hindering factors of implementing this approach.

## Data collection and management

All participants were monitored for possible TB symptoms, IPT related side effects, phone and in-person appointment keeping, and IPT adherence and completion. Existing data collection systems and tools such as the electronic medical records (EMR) and ART registers were adapted to capture and record project data from month 3–9 of implementation (September 2019-April 2020) (Fig 2). In May 2020, pilot staff trained in research ethics and participant confidentiality extracted participant demographics, reported IPT-related side effects, appointment keeping and IPT completion from the EMR. Data was checked for completeness, cleaned, and stored in a secured and password-protected PostgreSQL database for analysis. We de-identified the data and only included fields necessary for analysis.

## Data analysis

The primary outcome was self-reported IPT completion defined as completing the dispensed 6-month course of IPT. Secondary outcomes were reported side effects likely associated with IPT (burning sensation in fingers and toes; itchy skin, yellow eyes, tongue and palms; joint pains; nausea; vomiting; stomach pains and fever) and the percentage of scheduled phone appointments completed. We used descriptive statistics including frequencies and proportions to describe participant characteristics such as age, gender, time on ART and outcome variables. Fisher's exact test was used to determine if participant demographics were statistically associated with primary and secondary outcomes.

## Ethical consideration

The University of Zambia Biomedical Research Ethics Committee (UNZABREC) approved, and the National Health Research Authority (NHRA) gave authority to conduct the pilot. Since this evaluation was implemented within routine ART program, signed informed consent was not a requirement for participation.

## Results

### Participant characteristics

A total of 1167 PLHIV were screened, of whom 818 (70.1%) were eligible and enrolled within the 30-day period. Of the 349 (29.9%) excluded, 178 (15.2%) had no phone; 122 (10.5%) were already on IPT; 26 (2.2%) had recently completed IPT/TB treatment; 13 (1.1%) had TB symptoms; 7 (0.6%) were pregnant; and 3 (0.3%) declined to participate in the pilot (Fig 4). The 818 participants had median age of 42 years (IQR = 35, 49), and 540 (66%) were females. Over half (56.6%) of the participants were on ART for at least 5 years (Table 1).

### IPT completion

Of the 818 enrolled participants, 738 (90.2%) reported completing the full 6-month course of IPT; there were no significant differences in completion rate by age, sex, or duration on ART (Table 1). Of the 80 (9.8%) of participants who did not complete the full course of IPT, 45 (5.5%) initiated IPT and then stopped due to medical advice as they experienced side effects,

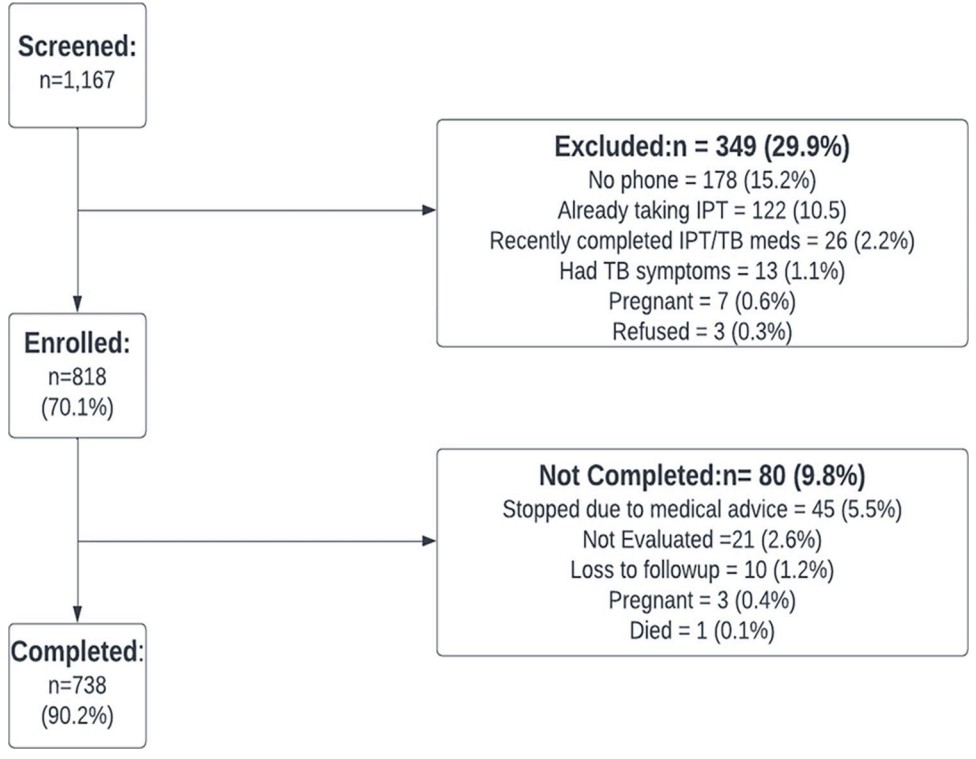

**Fig 4. Enrollment flow chart.**

21 (2.6%) were not evaluated, 10(1.2%) were lost to follow-up and 3(0.4%) became pregnant (Fig 4). One participant died in week 1 of the study but had not initiated IPT according to their family.

## Keeping scheduled appointments

Five hundred and thirty-nine (65.9%) participants kept all seven phone appointments; there were insignificant differences in completion rate by age, sex, or duration on ART (Table 1). Appointment keeping was lower in the first three months of IPT initiation (Fig 5). Approximately 20% of participants did not keep their phone appointments at weeks 2, 4 and 8. Following two failed phone calls on two separate days a physical visit was then conducted, on average, 75% of these were reached physically and of these 95% reported adherence with IPT.

**Reported side effects.** Sixty-six participants (8.1%) reported side effects, with insignificant differences by age, sex, or duration on ART (Table 1). Reported side effects were frequent in the early weeks of IPT initiation (Fig 6). Whilst none of the side effects were classified as serious according to Zambia Medicines Regulatory Authority (ZAMRA) definition [23], forty-five (5.5%) of participants were advised by their healthcare workers to discontinue IPT for fear of side effects worsening during IPT administration (Fig 4).

## Lessons learned: Successes

**Effectively adapting existing processes and clinical tools.** Adapting existing process and clinical tools ensured identification of the appropriate facility staff to provide screening, counselling, enrolment, IPT initiation and follow-up services. Additionally, it harmonised the tools

**Table 1. Baseline characteristics and outcomes of enrolled participants.**

| Characteristics | Number of participants (% of Total) | Number (%) completed treatment | | | P-value | Number (%) Kept All Appointments | | P-value | Number (%) Experienced Side Effects | | | P-value |
|---|---|---|---|---|---|---|---|---|---|---|---|---|
| | | Yes | No | Missing | | Yes | No | | Yes | No | Missing | |
| Age (years): median (IQR) | 42, (35–49) | 42 (35–49) | | | | 42 (35–49) | | | 44.5 (38–49) | | | |
| <30 | 89 (10.9%) | 77 (88.5%) | 9 (10.1%) | 3 (3.4%) | 0.2 | 60 (67.4%) | 29 (32.6%) | 0.9 | 2 (2.3%) | 58 (65.2%) | 29 (32.6%) | 0.2 |
| 30–39 | 270 (33%) | 242 (89.6%) | 27 (10%) | 1 (0.4%) | | 174 (64.4%) | 96 (35.6%) | | 18 (6.7%) | 165 (61.1%) | 87 (32.2%) | |
| 40–49 | 286 (35%) | 263 (91.0%) | 22 (7.7%) | 1 (0.4%) | | 192 (67.1%) | 94 (32.9%) | | 30 (10.5%) | 172 (60.1%) | 84 (29.4%) | |
| 50+ | 173 (21.15%) | 156 (90.2%) | 16 (9.3%) | 1 (0.4%) | | 113 (65.3%) | 60 (34.7%) | | 16 (9.5%) | 104 (60.1%) | 53 (30.6%) | |
| **Sex** | | | | | 0.7 | | | 0.8 | | | | 0.5 |
| Female | 540 (66%) | 484 (89.6%) | 52 (9.6%) | 4 (0.7%) | | 354 (65.6%) | 186 (34.4%) | | 48 (8.9%) | 325 (60.2%) | 167 (30.93) | |
| Male | 278 (34%) | 254 (91.4%) | 22 (7.9%) | 2 (0.7%) | | 185 (66.6%) | 93 (33.4%) | | 18 (6.5%) | 174 (62.6%) | 86 (30.9%) | |
| **ART Duration (years)** | | | | | 0.2 | | | 0.5 | | | | 0.5 |
| 1–2 | 61 (7.5%) | 52 (85.3%) | 8 (13.1%) | 1 (1.6%) | | 37 (60.7%) | 24 (39.3%) | | 3 (5%) | 35 (57.4%) | 23 (37.7%) | |
| 3–5 | 153 (18.7%) | 139 (90.9%) | 12 (7.8%) | 2 (1.3%) | | 106 (69.3%) | 47 (30.7%) | | 10 (6.6%) | 97 (63.4%) | 46 (30.1%) | |
| 5+ years | 463 (56.6%) | 418 (90.3%) | 44 (9.5%) | 1 (0.2%) | | 299 (64.6%) | 164 (35.4%) | | 37 (8%) | 279 (60.26%) | 147 (31.6%) | |
| Missing | 141 (17.2%) | 129 (91.5%) | 10 (7.1%) | 2 (1.4%) | | 97 (68.8%) | 44 (31.2%) | | 16 (11.4%) | 88 (62.4%) | 37 (26.2%) | |
| **Total** | **818 (100%)** | **738 (90.2%)** | **74 (9.1%)** | **6 (0.7%)** | | **539 (65.9%)** | **279 (34.1%)** | | **66 (8.1%)** | **499 (61%)** | **253 (30.9%)** | |

(screening tools, clinical records and SOPs) thus minimized burden on facility staff and ensured that the intervention was not viewed as an additional responsibility.

**Demand creation and client empowerment.** CHWs reported that the strategy of utilizing structured (1) educational messages messages and (2) checklist and script empowered them to generate demand for the intervention, to allay participants' concerns and consequently empowered participants to monitor and take charge of their health.

**Cultivating a collaborative structured learning environment.** The monthly meetings at the facility cultivated an iterative and collaborative learning environment that promoted and provided a platform for initial and ongoing success of integrating IPT into FT models.

**Promoting ownership.** The HCWs showed absolute involvement and support for this pilot project, which enabled the implementation of this new approach.

**Ensuring commodity security.** Securing the supply chain system ensured constant availability of both INH and Pyridoxine (Vitamin B6).

## Lessons learned: Challenges

**Phone usage.** The intervention model included 7 phone check-ins over the 6-month intervention, meaning that access to phones was critical for pilot implementation and effectiveness. Although access to a phone was an eligibility criterion, approximately 20% of participants did not keep their appointments in week 2, months 1 and 2 as phones were either

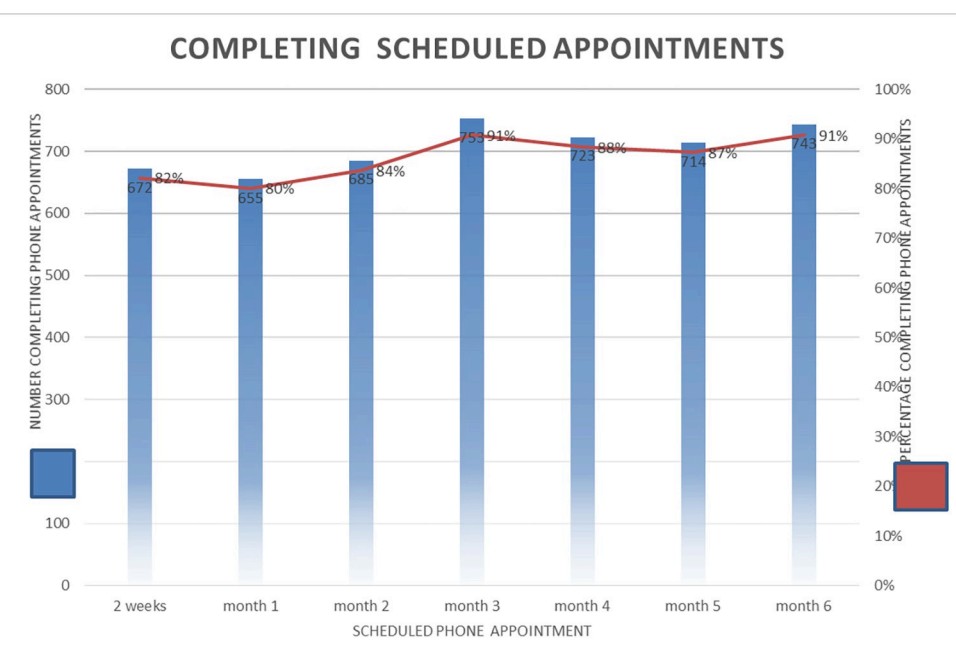

**Fig 5. Completion of phone visits by month.**

switched off or went unanswered despite repeated attempts. For those not reached by phone, CHWs conducted in-person follow ups to ascertain IPT adherence, about 75% were reached physically of whom 95% reported taking IPT as prescribed. During the in-person follow-ups, CHWs emphasized the importance of the phone follow-ups this improved phone usage in subsequent months (Fig 5).

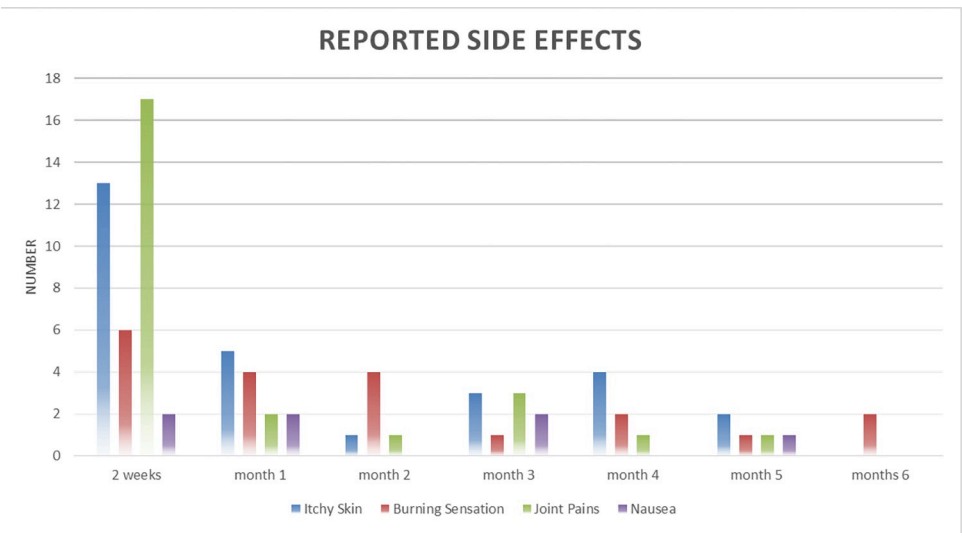

**Fig 6. Reported side effects.**

## Discussion

This pilot represents one of the first examples of integrating long refills of TPT (6MMSD) into a facility-based DSD model with utilization of phone calls at shorter intervals for continuous education, close monitoring and adherence support. We found that integration of TPT uptake amongst participants was considerably higher than the national average, and self-reported TPT completion rates were similar to those found nationwide. Our project builds on examples from case studies findings conducted in South Africa and the Democratic Republic of Congo that indicated integration of TPT into DSD models could be successfully adopted as a TB screening strategy and that longer refills of TPT were feasible [16]. Additionally, findings of high IPT completion rates and low reports of IPT side effects from this pilot are consistent with early findings from the implementation of IPT in DSD models in rural Uganda [24].

In this pilot project we found that 90.2% of the participants who accepted this innovation completed their IPT, similar to Zambia's reported national completion rate of (90%) [25]. Although design of this study did not include control or comparison groups, this high completion rate is encouraging and may have been due to the combination of program efficiency and access to ongoing phone-based counselling and support, which we believe is an innovative strategy of facilitating social support. Studies by Grimsrud *et al*. and Nachenga *et al*. underscore the significant contribution of social support in promoting IPT completion in community-based HIV DSD models [26, 27]. Further observations made by authors from South Africa and Uganda, where higher ITP completion rates were reported in clients exposed to enhanced social support in both DSD community-based models and standard of care [24]. Whilst our pilot integrated IPT in a facility-based model, we believe that the continuous social support provided via phone promoted treatment completion.

Furthermore, high completion rates seen in our pilot are concordant with evidence from a quality improvement collaborative in Uganda that showed 89% completion rates and two studies conducted in Swaziland and Uganda that showed improvement of TPT completion in models that integrated TB and HIV services [24, 28, 29]. In Swaziland, IPT completion rates in facility based (FB) models and community based (CB) models was reported at 89% and 91% respectively. Both models utilized trained HCWs to provide support and counselling [29]. Similarly, the Ugandan cross-sectional study showed a higher completion rate in people in DSD models (72%) than in routine care (53%). Results from this study suggested that the DSD models allowed for stronger HCW- client relationship, education, communication and empowerment [24].

Uptake and coverage of IPT amongst PLHIV in Zambia ranges from 18% to 49% while in this pilot 70.1% of all enrolled participants accepted the intervention [4, 30]. We hypothesize that the person-centered and convenient model being piloted led to higher than usual 70.1% uptake rate. DSD models that extended clinic visits have shown similar uptake success in HIV programs thus, supporting clients' preference for extended integrated visits [31, 32].

Given the push to leverage DSD models to scale up TPT for PLHIV it was important to evaluate TPT in routine care, particularly in relation to side effects and treatment completion [33]. Collecting and evaluating side effects data will capacitate the HCWs to reasonably inform patients regarding the benefits versus risks of TPT [34]. HCWs have often conveyed fears of IPT medication toxicity due to concomitant use with ART [35, 36]. Our evaluation data showed that at 6 months of IPT initiation only about 8.1% of the participants experienced side effects thus suggesting the fears to be unfounded. These findings are corroborated by the 2019 Centre for Global Health Division TPT myth and fact, which concluded that risk of adverse events in patients on TPT was low, with <10% side effects recorded from various studies [36]. Additionally the national QI collaborative in Uganda similarly reported that <10% of the study participants experienced INH adverse events [28].

This proof-of-concept pilot shows the importance of utilizing existing processes, establishing HCWs project ownership, promoting client empowerment and fostering iterative and collaborative learning environment all of which were pivotal to the success. A 2019 debate of lessons learnt during IPT implementation in Zambia closely linked programmatic issues such as HCWs capacity to manage patients, limited demand for TPT services, drug stock outs and patient IPT knowledge to the IPT completion and scale up [22]. A weak supply chain is a significant hinderance to treatment completion and TPT scale up [11]. In the prospective cohort of improving IPT delivery in DSD models in Swaziland, erratic IPT supply was one of the reasons for non-completion of TPT [29]. Therefore, securing the supply chain was useful for implementation and maintenance of this pilot.

Strengths of this program evaluation include the innovative pilot design that does not only reduce the burden of patients frequenting the health facility, but moreover promotes MoH recommendation of minimizing PLHIV side effects risks by ensuring exhaustive education of the patients and close monitoring [2]. Additionally provision of IPT long refills and utilization of phone calls in between clinic visits is an intervention we believe may be leveraged to support uptake and completion of other TPT regimens such as Isoniazid plus Rifampicin (3HP) and Isoniazid plus Rifampicin (3HR) in Zambia [2]. Another strength of this pilot was the documentation of the implementation process. Limitations include those typically associated with a single-site proof-of-concept project of this kind: the absence of a control or comparison group other than historical experience with IPT in Zambia, lack of TB infection outcomes reliance on self-report to assess IPT completion, the purposive selection of one urban health facility and the exclusion of PLHIV without phones, all of which limit the strength and generalizability of our results. The description of lessons learnt draw on the experiences of the project team and while we believe it is important to share these experiences, we also recognize the subjective nature of the successes and challenges reported. Nonetheless, we are confident that the results of this pilot project will contribute to the growing body of literature on the importance of integrating TPT with other health services into DSD models for people with HIV [37]. We recommend that future studies utilize a structured thematic/realist analysis to synthesize key lessons learned and objective alternative approaches to measure adherence.

## Conclusions

This pilot demonstrated that an integrated MMSD/IPT model involving synchronized IPT and HIV clinical visits, dispensing 6 months of IPT and ART, and delivering adherence counseling and follow up via phone was associated with excellent IPT uptake and completion rates. This promising public health strategy is worth further study and may improve coverage and completion of TPT amongst PLHIV, especially in countries that have taken DSD to scale and in which mobile phone penetration is high. Fostering project ownership, empowering clients, building health care worker capacity, collaboration with key stakeholders, mobilizing human resources and attending to supply chain security, were essential to successful pilot implementation.

## Supporting information

**S1 Table. Sensitivity analysis of participants who kept/did not keep all phone appointments and participants with /without phones.**
(DOCX)

**S1 Data. Anonymized study data.**
(DTA)

**S2 Data. Anonymized study data.**
(DO)

**S1 Text. Study forms and tools.**
(PDF)

**S2 Text. Study forms and tools.**
(PDF)

**S3 Text. Study forms and tools.**
(PDF)

## Acknowledgments

We are indebted to the Government of the Republic of Zambia, through the Ministry of Health, for their valuable guidance and support during implementation. We would like to thank the DSD Task Force and the TB National Program for leading the conceptualization and providing meaningful guidance throughout. We are grateful to ICAP at Columbia University and the CQUIN Learning Network for this great initiative that culminated into novel findings. Lastly, we thank and salute the participants and health care workers for their efforts and contributions to the larger vision of achieving epidemic control.

## Author Contributions

**Conceptualization:** Mpande Mukumbwa-Mwenechanya, Muhau Mubiana, Khozya Zyambo, Maureen Simwenda, Nancy Zongwe, Linah Kampilimba Mwango, Felton Mpesela, Fred Chungu, Felix Mwanza, Khunga Morton, Priscilla Lumano Mulenga.

**Data curation:** Paul Somwe, Estella Kalunkumya.

**Formal analysis:** Paul Somwe, Samuel Bosomprah.

**Methodology:** Priscilla Lumano Mulenga.

**Project administration:** Mpande Mukumbwa-Mwenechanya, Muhau Mubiana, Estella Kalunkumya.

**Software:** Paul Somwe.

**Supervision:** Muhau Mubiana, Khozya Zyambo, Carolyn Bolton-Moore.

**Validation:** Anjali Sharma.

**Writing – original draft:** Mpande Mukumbwa-Mwenechanya.

**Writing – review & editing:** Mpande Mukumbwa-Mwenechanya, Muhau Mubiana, Khozya Zyambo, Maureen Simwenda, Nancy Zongwe, Linah Kampilimba Mwango, Miriam Rabkin, Felton Mpesela, Fred Chungu, Felix Mwanza, Peter Preko, Carolyn Bolton-Moore, Samuel Bosomprah, Anjali Sharma, Khunga Morton, Prisca Kasonde, Lloyd Mulenga, Patrick Lingu, Priscilla Lumano Mulenga.

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
