## [Decision Letter · Decision Letter 0]

17 Nov 2022

PGPH-D-22-01115

Integrating Isoniazid Preventive Therapy into the Fast-Track HIV Treatment Model in Urban Zambia: A Proof-of -Concept Pilot Project

Dear Dr. Mukumbwa-Mwenechanya,

Thank you for submitting your manuscript to PLOS Global Public Health. After careful consideration, we feel that it has merit but does not fully meet PLOS Global Public Health’s publication criteria as it currently stands. Therefore, we invite you to submit a revised version of the manuscript that addresses the points raised during the review process.

Thank you for your patience in our evaluation of your paper which addresses an important aspect of TB/HIV integration. Below are the comments of the reviewers, with which I agree. We look forward to receiving a revised verison. A few comments need to be addressed before publication.

We look forward to receiving your revised manuscript.

Kind regards,

Sabine Hermans

Academic Editor

Journal Requirements:

a. State what role the funders took in the study. If the funders had no role in your study, please state: “The funders had no role in study design, data collection and analysis, decision to publish, or preparation of the manuscript.”

b. If any authors received a salary from any of your funders, please state which authors and which funders.

2. In the online submission form, you indicated that "The Government of Zambia allows data sharing when applicable local conditions are satisfied. In this case, the data from the study will be made available to any interested researchers upon request. The CIDRZ Ethics and Compliance Committee is responsible for approving such request. To request data access, one must write to the Committee chair/Chief Scientific Officer, Dr. Roma Chilengi, (Roma.Chilengi@cidrz.org) or the Secretary to the Committee/Head of Research Operations, Ms. Hope Mwanyungwi (Hope.Mwanyungwi@cidrz.org) mentioning the intended use for the data. The Committee will then facilitate review and authorization to release the data as requested. Data requests must include contact information, a research project title, and a description of the intended use.". All PLOS journals now require all data underlying the findings described in their manuscript to be freely available to other researchers, either 1. In a public repository, 2. Within the manuscript itself, or 3. Uploaded as supplementary information.

Additional Editor Comments (if provided):

Reviewers' comments:

Reviewer's Responses to Questions

**Comments to the Author**

1. Does this manuscript meet PLOS Global Public Health’s publication criteria? Is the manuscript technically sound, and do the data support the conclusions? The manuscript must describe methodologically and ethically rigorous research with conclusions that are appropriately drawn based on the data presented.

Reviewer #1: Partly

Reviewer #2: Yes

2. Has the statistical analysis been performed appropriately and rigorously?

Reviewer #1: Yes

Reviewer #2: No

3. Have the authors made all data underlying the findings in their manuscript fully available (please refer to the Data Availability Statement at the start of the manuscript PDF file)?

Reviewer #1: No

Reviewer #2: No

4. Is the manuscript presented in an intelligible fashion and written in standard English?

Reviewer #1: Yes

Reviewer #2: Yes

5. Review Comments to the Author

Reviewer #1: Major concerns

1. A major concern is the way in which the lessons learnt and challenges were captured. More information is needed on whether a structured analytical approach was followed, for instance a thematic analysis to “… synthesize key lessons learned…” or in which alternative way the “… implementers, facility-based clinicians and community health workers assessed the tools and processes to be effective and relatively efficient.” (lines 203-233)

2. Second, adherence and side effects were measured through self-report. Did the authors consider an alternative approach to measure adherence, for instance 99DOTS or measuring drug levels through a urinary assay? Self-reported adherence is notoriously overestimated. The authors might want to include this as a suggestion for future studies. For side effects, a telemedicine approach might be valuable.

3. Third, did the authors consider a sensitivity analysis to look at the potential participants without phones who were excluded (15%). A selection bias might have been introduced if these potential participants were substantially different from their peers, which might have been the case particularly on a socio-economic level if they did not have phones. It would also be interesting to know whether the participants who could not be found at home visits (25%), were different from their peers.

4. Finally, the reported IPT adherence in the project was 90.2% vs. the national completion rate of 90%. Since there is no apparent statistical significance between 90.2% and 90%, what would the added value be when this pilot is scaled-up? I would suggest an additional emphasis on uptake of IPT, which was significantly higher than in the standard of care (>70% vs. <50%).

Minor concerns

1. Line 84: please clarify this sentence “We believed this site to be relatively common in our country (even if not typical) to quickly learn and generate valuable lessons…” – my assumption is that the authors wanted to make the point that the site experience could be generalized in a way.

2. Please clarify in Figure 3, if a HIV clinic visit/ART pick-up was conducted at three months anyway, why would participants still need to be called for IPT? Should the HIV clinic visit not be scheduled at 6 months?

3. Please check Table 1, the last column does not add up (Sex and ART duration).

Reviewer #2: General comment: Relevant topic regarding delivery of TB Preventive Therapy (TPT) in the era of Differentiated Service Delivery (DSD).

1. One of the most worried things in TPT is Breakthrough TB diseases. Hence, WHO recommends regular screening of TB using symptoms and screening. There is no TB screening in the piloted model. Was it done?

I think for the model to function well, there should be a way to monitor development of TB during TPT. Continuation of TPT in the presence of TB diseases leads to inappropriate treatment which may lead to the development of drug resistance mutation

2. The authors mentioned about using Fishers exact test to determine the association between outcomes and exposures. However, there were not 2 by 2 shown despite stating that there were no statistically significant differences.

By virtue of biological plausibility or previous literature, one would go further to carry multivariable regression analysis to determine the relationship.

6. PLOS authors have the option to publish the peer review history of their article (what does this mean?). If published, this will include your full peer review and any attached files.

**Do you want your identity to be public for this peer review?** For information about this choice, including consent withdrawal, please see our Privacy Policy.

Reviewer #1: No

Reviewer #2: No

---

## [Editor Report · Decision Letter 1]

2 Feb 2023

Integrating Isoniazid Preventive Therapy into the Fast-Track HIV Treatment Model in Urban Zambia: A Proof-of -Concept Pilot Project

PGPH-D-22-01115R1

Dear Dr Mukumbwa-Mwenechanya,

We are pleased to inform you that your manuscript 'Integrating Isoniazid Preventive Therapy into the Fast-Track HIV Treatment Model in Urban Zambia: A Proof-of -Concept Pilot Project' has been provisionally accepted for publication in PLOS Global Public Health.

Best regards,

Sabine Hermans

Academic Editor
